# Immune Checkpoint Blockades in Triple-Negative Breast Cancer: Current State and Molecular Mechanisms of Resistance

**DOI:** 10.3390/biomedicines10051130

**Published:** 2022-05-13

**Authors:** Hyungjoo Kim, Je-Min Choi, Kyung-min Lee

**Affiliations:** 1Department of Life Science, College of Natural Sciences, Hanyang University, Seoul 04763, Korea; hjook1990@hanyang.ac.kr; 2Penta Medix Co., Ltd., Seongnam-si 13449, Korea; 3Research Institute for Natural Sciences, Hanyang University, Seoul 04763, Korea; 4Research Institute for Convergence of Basic Sciences, Hanyang University, Seoul 04763, Korea; 5Hanyang Institute of Bioscience and Biotechnology, Hanyang University, Seoul 04763, Korea

**Keywords:** immune checkpoint blockade, TNBC, resistance

## Abstract

Immune checkpoint blockades (ICBs) have revolutionized cancer treatment. Recent studies have revealed a subset of triple-negative breast cancer (TNBC) to be considered as an immunogenic breast cancer subtype. Characteristics of TNBC, such as higher mutation rates and number of tumor-infiltrating immune cells, render the immunogenic phenotypes. Consequently, TNBCs have shown durable responses to ICBs such as atezolizumab and pembrolizumab in clinic. However, a significant number of TNBC patients do not benefit from these therapies, and mechanisms of resistance are poorly understood. Here, we review biomarkers that predict the responsiveness of TNBCs to ICB and recent advances in delineating molecular mechanisms of resistance to ICBs.

## 1. Introduction

Breast cancer is one of the most commonly diagnosed cancers worldwide, accounting for 685,000 deaths in 2020 [1]. Molecular classification of breast cancer into subtypes based on expression of estrogen receptor (ER), progesterone receptor (PR), and human epidermal growth factor receptor 2 (HER2) have provided therapeutic options that have improved clinical outcome. These receptors can serve as both prognostic and predictive markers for targeted therapies. However, breast cancer patients with triple-negative breast cancer (TNBC), which are known to lack expression of ER, PR, and HER2, currently have limited standard treatment option with a targeted therapy. In addition, while early breast cancer is considered curable, metastatic breast cancer is considered treatable but not curable with current treatment strategies. Therefore, there is a need for innovative therapeutic options to improve cancer patient outcome.

Immunotherapeutic approaches against cancer, including immune checkpoint blockades (ICBs) and chimeric antigen receptor (CAR) T-cell therapies, have been proven to be effective in generating promising results in various types of cancers [2,3,4,5]. Diverse factors derived from the tumor itself and from the environment surrounding the tumor can influence the immune context of a tumor. These factors including the mutational load, rates of neoantigens, and the presence and diversity of tumor-infiltrating lymphocytes (TIL) have been shown to dictate the susceptibility of a tumor to immunotherapy [6].

Traditionally, breast cancer has been shown to be immunologically silent; therefore, it is unlikely to benefit from immunotherapy [7]. However, compared to ER-positive breast cancer intrinsically exhibiting lower immunogenicity, HER2-positive breast cancer and TNBC are known to be immunogenic [8]. Among breast cancer subtypes, HER2+ and TNBC subtypes are known to likely have a larger number of TILs compared to the luminal subtype [9]. In addition, high levels of TILs were associated with better prognosis in early stage TNBC and HER2+ breast cancer [10]. A study that utilized the TCGA dataset to analyze the association between genomic alterations and immune metagene expression revealed that the higher immune gene expression and lower clonal diversity indicates the immune pruning effect in TNBC and HER2+ breast cancers [11]. Especially, TNBC is thought to be the most immunogenic subtype of breast cancers due to its genomic instability and higher mutational burden [12]. Liu and colleagues, through performing a genomic meta-analysis of immunogenic signatures using large-scale breast cancer datasets, demonstrated that TNBCs are more immunogenic than other breast cancer subtypes [13]. Moreover, a study utilizing a bioinformatical pipeline to analyze HLA class I restricted neoepitopes in the TCGA dataset demonstrated that the total mutational burden was higher in TNBC than in other breast cancer subtypes [14]. In line with the immunogenic features, cancer immunotherapy has shown profound efficacy to prolong patient survival in clinical trials, eventually leading to approval of pembrolizumab, a programmed cell death-1 (PD-1) inhibitor, for the treatment of TNBC patients (ClinicalTrials.gov Identifier: NCT03036488, last visited on 22 October 2021). However, the use of immunotherapy is still restricted in a sizable number of TNBC patients due to a limitation in adequate predictive and prognostic biomarkers.

In this review, we present immunotherapies that are currently available or potentially approved upon ongoing clinical trials in TNBCs, and we introduce putative molecular mechanisms for resistance to immunotherapy, which have been demonstrated by laboratory-based studies.

## 2. Immune Checkpoint Blockade Clinical Trials in TNBC

### 2.1. Pembrolizumab

In the KEYNOTE-012 Phase 1b study (NCT01848834), the safety and anti-cancer activity of pembrolizumab, a humanized monoclonal antibody specific for PD-1, was evaluated in patients with advanced TNBC, urothelial cancer, head and neck cancer, and gastric cancer [15]. The overall response rate (ORR) was 18.5% and median time to response was 17.9 weeks. Next, cohort A of the KEYNOTE-086 Phase 2 study (NCT02447003) examined the efficacy and safety of pembrolizumab in 170 patients with mTNBC who have previously received systemic treatment [16]. Among these patients, 105 (61.85%) patients were screened to have programmed cell death ligand 1 (PD-L1)-positive tumors. The ORR was 5.3% in the total, 5.7% in the PD-L1-positive, and 4.7% in the PD-L1 negative population, respectively. The median progression-free survival (PFS) and overall survival (OS) were 2.0 and 9.0 months, respectively. In the cohort B of the KEYNOTE-086 Phase 2 study (NCT02447003), mTNBC patients with no prior systemic treatment were enrolled for the study [17]. The ORR was 21.4% with a median PFS and OS of 2.1 and 18.0 months, respectively. The median time to response and duration of response was 2.0 and 10.4 months, respectively. The findings from cohort A and B of the KEYNOTE-086 study suggested that pembrolizumab shows higher response rates in PD-L1-positive mTNBC patients in earlier lines of treatment. In a following Phase 3 clinical trial, KEYNOTE-119 (NCT02555657), the efficacy of pembrolizumab and chemotherapy was compared in previously treated mTNBC patients [18]. The ORR of the pembrolizumab-treated group and chemotherapy-treated group was 9.6% and 10.6%, respectively. There was no significant difference in the median OS between those groups. However, patients with higher PD-L1 expression exhibited longer median OS and increased response duration in the pembrolizumab-treated group.

Various chemotherapeutic agents have been shown to potentially elicit immunogenic responses by inducing MHC expression, increasing tumor antigens, upregulating co-stimulatory molecules, and downregulating co-inhibitory molecules [19]. In that context, many clinical trials have been investigating the effect of combinatorial treatment regimens of PD-1/PD-L1 inhibitors with chemotherapy in TNBC and have shown encouraging results (Table 1). In the phase 3 KEYNOTE-355 trial (NCT02819518), the efficacy of pembrolizumab combined with chemotherapy was tested in mTNBC patients. Among the 847 patients, 566 patients and 281 patients were randomly allocated to the pembrolizumab chemotherapy group and placebo chemotherapy group, respectively. In the PD-L1-positive mTNBC, patients with a combined positive score of 10 or more, median PFS was 4.1 months longer in the pembrolizumab chemotherapy patients compared to the placebo-chemotherapy patients (9.7 months vs. 5.6 months) [20]. Moreover, cisplatin and doxorubicin treatment led to increased expression of immune-related genes, suggesting a proinflammatory role of chemotherapeutic reagents. In the phase 1b/2 KEYNOTE-150/ENHANCE-1 study (NCT02513472), in which mTNBC patients were treated with eribulin mesylate in combination with pembrolizumab, the ORR and PFS was 23.4% and 4.1 months, respectively. In the phase 2 I-SPY2 study (NCT01042379), 181 early-stage breast cancer patients were assigned into the control group receiving standard chemotherapy, while 69 patients received standard chemotherapy with pembrolizumab [21]. The pathologic complete response (pCR) rates were 60% vs. 22% for the chemotherapy plus pembrolizumab group vs. chemotherapy group in TNBC patients [21]. In the phase 1b KEYNOTE-173 study (NCT02622074), six different combinatorial treatment regimens of pembrolizumab plus chemotherapy showed an overall pCR rate of 60%, ranging from 49% to 71% [22]. The phase 3 KEYNOTE-522 trial (NCT03036488) demonstrated that the pCR rates were 64.8% and 51.2% in the pembrolizumab plus chemotherapy and pembrolizumab plus placebo group, respectively. Consequently, the United States Food and Drug Administration (FDA) approved pembrolizumab in combination with chemotherapy as neoadjuvant treatment in 2021.

### 2.2. Atezolizumab

In a phase I study testing the safety and efficacy of single agent atezolizumab, a monoclonal antibody against PD-L1, in patients with advanced solid and hematologic malignant tumors (NCT01375842), the ORR was 10% in 116 mTNBC patients [23]. Subgroup analysis revealed that patients who received atezolizumab as first line of treatment showed higher ORR of 24% compared to those who received it in second line settings and beyond having ORR of 6%. This study demonstrated that the efficacy of single agent atezolizumab was associated with higher amounts of PD-L1-positive immune cells. Conversely, patients with high tumor burden, liver metastases, and/or increased lactate dehydrogenase levels had decreased clinical benefits in the study. The phase 3 IMpassion 130 study (NCT02425891) demonstrated that the median OS was 21.0 months in mTNBC patients who received atezolizumab plus nab-paclitaxel, while it was 18.7 months in the patients who received placebo plus nab-paclitaxel [24]. In PD-L1-positive patients, the OS improvement was 7 months with an OS of 25.0 months and 18.0 months in the atezolizumab group and placebo group, respectively. However, the following phase 3 IMpassion 131 study (NCT03125902) could not find significant differences in primary PFS or final OS between the atezolizumab group and placebo group [25]. The phase 3 IMpassion 132 study (NCT03371017) is being conducted to evaluate the efficacy of atezolizumab plus chemotherapy in inoperable recurrent TNBCs.

### 2.3. Other PD-1/PD-L1 Inhibitors

The phase 2 TONIC study (NCT02499367) investigated the effectiveness of nivolumab, a humanized anti-PD-1 monoclonal antibody, in various treatment regimens. In this study, cisplatin and doxorubicin pretreatment followed by nivolumab resulted in an increased ORR of 23% and 35%, respectively, compared to 17% in mTNBC patients without pretreatment [26]. The JAVELIN Solid Tumor study (NCT01772004), which treated metastatic breast cancer patients with avelumab, a human antibody targeting PD-L1, reported ORR of 3.0% in the total population and 5.2% in the TNBC subgroup [27]. In addition, the A-Brave, a phase 3 randomized adjuvant study, is evaluating the efficacy of avelumab in TNBC patients (NCT02926196). The phase 3 GeparNuevo clinical trial (NCT02685059) evaluated the efficacy of combination therapy of durvalumab, an anti-PD-L1 monoclonal antibody, with chemotherapy in primary TNBC [28]. Patients were randomized into groups receiving nab-paclitaxel with durvalumab or placebo treatment followed by epirubicin/cyclophosphamide. The pCR rate was 53.4% in the durvalumab group, while it was 44.2% in the placebo group [28].

### 2.4. Anti-CTLA-4 Inhibitors

In a pilot study to evaluate the effects of preoperative ipilimumab, an anti-CTLA-4 monoclonal antibody, in combination with cryoablation, 19 breast cancer patients selected across subtypes were treated with cryoablation, ipilimumab, or both [29]. The rationale for this study was that the tumor cryoablation process could aid tumor-specific immunity by inducing cell lysis, which may enhance tumor antigen presentation to the immune cells [30]. The authors demonstrated that treatment with the cyro-immunotherapy increased not only Ki67+ T cells but also the ratio of Ki67+ effector T cell to Ki67+ regulatory T cell compared to cyroablation or ipilimumab alone [29]. Another study investigated the activity of tremelimumab, a monoclonal antibody targeting CTLA-4, in combination with exemestane in hormone receptor (HR)-positive metastatic breast cancer patients [31]. Among the total of 26 patients, 11 patients (42%) had stable disease for ≥12 weeks [31]. Furthermore, the number of inducible costimulator-expressing T cells were increased, while Foxp3+ Tregs were decreased following treatment [31].

### 2.5. Combinational Treatments

Poly (ADP-ribose) polymerase inhibitors (PARPi), such as olaparib, niraparib, and talazoparib, inhibit single-strand DNA break repair, thus leading to synthetic lethality in homologous recombination repair-defective cancer [32]. Olaparib and talazoparib have been approved for breast cancer patients with *BRCA* germline mutation. By inducing impaired DNA repair, PARPi was shown to increase neoantigens and tumor mutational burden, both of which predict significant response to ICB treatment [33]. PARPi was also shown to upregulate PD-L1 expression in breast cancer through GSK3β inactivation, further providing rationale for combinatorial treatment of PARPi with ICBs [34]. Given its immunogenic potency, several trials are ongoing to test the efficacy of PARPi in combination with ICBs in mTNBC patients (NCT02657889, NCT03167619).

Cyclin dependent kinases (CDKs) are master regulators that are involved in cell cycle transition and cell division, required for cancer initiation and progression [35,36]. CDK4/6 inhibitors such as palbociclib, ribociclib, and abemaciclib have been approved by the FDA for treatment of HR-positive metastatic breast cancer [35]. Of note, Goel et al. (2017) demonstrated that CDK4/6 inhibitors increase anti-tumor immunity by activating tumor antigen presentation and reducing proliferation of Tregs through suppressing DNA methyltransferase 1 activity [36]. Recent studies have also shown that combination treatment of CDK4/6 inhibitors with ICBs exhibits a synergistic effect on tumor control, suggesting that combining these therapies is a remarkable strategy for expanding ICB utility [37,38].

In addition to strategies combining molecular-targeted therapies with ICBs, other strategies include combination treatment of peptide vaccines or natural killer (NK) cells with ICBs. In a phase 1a/1b study (NCT03289962), the effects of autogene cevumeran, a personalized neoantigen specific vaccine, in combination with atezolizumab, is being explored. In addition, a phase 1b clinical study with mTNBC patients (NCT03362060) is exploring the efficacy of pembrolizumab in combination with PVX-410, a multi-peptide cancer vaccine composed of four synthetic peptides that are known to induce T cell-mediated immune response. The success in ex vivo expansion of patient-derived autologous and genetically engineered NK cells contributed to initiation of trails testing a NK cell-based regimen [39,40]. A phase 1 study (NCT04551885) is currently exploring the efficacy of FT-516, an engineered NK cell with a high affinity non-cleavable CD16 Fc receptor, with avelumab in various types of tumors including TNBC. Moreover, a phase 1b/2 study (NCT03387085) is evaluating the efficacy of high affinity NK (haNK) cells in combination with avelumab and other treatments including peptide vaccines, chemotherapeutics, and radiation therapy. Interim analysis revealed an encouraging result of 56% ORR and 78% disease control rate in nine mTNBC patients [41].

## 3. Predictive Biomarkers of Immunotherapy in TNBCs

Although ICBs have been approved to be used as therapeutic options in various types of cancers including TNBC, many patients have failed to benefit from it [42]. Therefore, a necessity exists in identifying predictive biomarkers that can guide decisions for patient selection. In addition, development of those predictive biomarkers could maximize clinical benefits while minimizing adverse effects by providing mechanistic rationale for appropriate immunotherapy treatment regimen selection.

### 3.1. Tumor Mutational Burden and Neoantigens

Tumor-associated antigens are composed of nonmutated self-antigens and neoantigens derived from nonsynonymous somatic mutations. Self-antigens include cancer-testis antigens such as the melanoma-associated antigens and New York esophageal squamous cell carcinoma 1, which are nonmutated proteins with restricted expression in the male germ cells normally but are expressed in cancer cells due to transcriptional reprogramming and epigenetic changes [43,44]. However, since the immune responses elicited by self-antigens are limited due to central tolerance, there are only few associations reported between self-antigen expression and increased ICB effectiveness. Conversely, numerous reports have demonstrated the association between somatic mutations, which give rise to tumor-specific neoantigens, and influence the response to ICBs in various types of cancers including non-small cell lung cancer, melanoma, and urothelial carcinoma [42]. Meta-analysis of 27 types of tumors revealed that high mutational burden was correlated with improved ORR in patients who received anti-PD-1/PD-L1 therapy [45]. However, although high mutational burden is associated with favorable outcomes in TNBCs due to its immunogenicity, the relatively low mutational load in breast cancer may explain why mutational burden might be invalid as a predictive biomarker in breast cancer [46,47].

### 3.2. PD-L1 Expression

Positive correlations between PD-L1 expression and response to ICBs targeting the PD-1/PD-L1 axis have been reported in various types of cancers [42]. Therefore, the FDA has approved immunohistochemical testing of PD-L1 expression as a companion diagnostic test, which aids in decision-making processes about the use of PD-1 or PD-L1 blockades, in various types of tumors [48]. However, some studies have shown no correlation between PD-L1 expression and ICB response [42]. Furthermore, a high proportion of PD-L1-negative patients have shown durable response to anti-PD-1 or anti-PD-L1 therapies [49]. The likely reasons for discrepancies in those studies may include the use of different detection platforms, antibody clones, and cut-off values for evaluating PD-L1 positivity. Routinely, five different antibody clones, namely, 22C3, 28-8, SP142, SP263, and 73-10, are used in diagnostic testing for pembrolizumab, nivolumab, atezolizumab, durvalumab, and avelumab, respectively [48]. The FDA has approved the PD-L1 IHC 22C3 pharmDx as a companion diagnostic test for identifying patients eligible for the immunotherapy regimen, pembrolizumab plus chemotherapy. Studies comparing clinical performance of the different antibody clones in TNBC demonstrated that the clones SP263 and 22C3 tend to show increased PD-L1 staining compared to SP142 [50,51,52]. In addition, a study that sought to assess the diagnostic concordance and reproducibility of PD-L1 staining assays among 19 pathologists revealed that TNBC sections stained with SP263 displayed elevated PD-L1-positivity compared to sections stained with SP142 [53]. These findings suggest that standardized and optimized methods examining PD-L1 expression should be required for guaranteeing PD-L1-positivity as a valid biomarker.

### 3.3. Tumor-Infiltrating Lymphocytes

ICB treatments such as anti-PD1/PD-L1 have been known to act at least partially by rejuvenating the preexisting anti-tumor immunity [42]. A high proportion of TILs have been shown to be associated with improved patient outcome in various types of cancer including melanoma, colorectal cancer, and TNBC [54,55]. In melanoma, spatio-temporal analysis of CD8+ T cells within the tumor using immunohistochemical analysis demonstrated that high densities of CD8+ T cells at the invasive margin were associated with a durable response to PD-1 blockade therapy [56]. In the KEYNOTE-086 trial (NCT02447003), mTNBC patients with higher stromal TILs (sTILs) showed better ORR in response to pembrolizumab [47]. A recent study, through performing immunogenomic analyses using multiple clinical datasets, demonstrated that 9p21 loss was associated with primary resistance to immunotherapy in metastatic urothelial cancer and advanced non-small-cell lung cancer [57]. Tumors with loss of 9p21 had lower TIL densities, altered spatial patterns, and compositions of TILs, and impaired antigen presentation and IFN-γ signaling. This immunologically “cold” phenotype was due to the downregulation of cytokines related to recruitment, activation, and clonal expansion of immune cells and upregulation of immunosuppressive genes [57]. The authors proposed that a loss of 9p21 could serve as a predictive biomarker in excluding patients who would not respond to ICB treatment, outperforming tumor mutational burden and PD-L1 positivity [57]. About 10% of TNBCs also harbor the homozygous deletion of 9p21 [58], encouraging preclinical and clinical studies, which assess the potential role of 9p21 loss as a predictive biomarker.

## 4. Putative Molecular Mechanisms of Resistance to ICB in TNBCs

One of the advantages of immunotherapy compared to conventional treatment options is the durability of the treatment effect. However, only a small proportion of patients could benefit from the long-lasting effect of immunotherapy due to low response rates and resistance [59]. The mechanisms that lead to resistance can be driven by tumor-extrinsic or -intrinsic factors (Figure 1). Tumor-extrinsic mechanisms are typically associated with a tumor immune microenvironment, whereas tumor-intrinsic mechanisms are promoted by more diverse factors including aberrant oncogenic signals, perturbated immune checkpoints, and inflammatory or immunosuppressive cytokines.

### 4.1. Tumor-Extrinsic Mechanisms of Resistance

Tumor-extrinsic factors include various compositions of the tumor microenvironment (TME) including effector T cells, regulatory T cells, myeloid-derived suppressor cells (MDSCs), tumor-associated macrophages (TAMs), et cetera. The majority of resistance mechanisms has been associated with tumor-infiltrating T cells. Sceneay at al. (2019) reported that a concomitant dysfunction of T cells with increased age limits the effects of ICB treatment in TNBC via an impairment in IFN signaling and antigen presentation machinery [60]. In addition, Tcf1+/PD-1+ stem-like CD8+ TILs contribute to the expansion of differentiated T cells within the tumor, in response to ICB treatment, suggesting a putative role of these subpopulations as a novel predictive biomarker [61]. Stem-like CD8+ T cells that typically express high levels of TCF1 have been found within the TIL population of various cancer types and are shown to produce terminally differentiated CD8+ T cells, thus constructing an intratumoral niche vulnerable to anti-tumor immunity [62]. In TNBC, cyclophosphamide and vinorelbine, which both activate stem-like CD8+ T cells, improved the efficacy of PD-1 blockades, implying a role of stem-like CD8+ T cells in response to ICBs [63]. Zhang and colleagues reported that abundant CXCL13+ T cells predict better clinical outcome of atezolizumab combined with paclitaxel in advanced TNBC [64]. Paclitaxel monotherapy attenuated the expansion of anti-tumor immune cells but upregulated immunosuppressive macrophages, suggesting that paclitaxel may not aid the effects of atezolizumab treatment [64]. The neutrophil-enriched subtype (NES) of TNBCs was identified as an ICB-resistant subtype due to the accumulation of immunosuppressive neutrophils in the TME [65]. Immunosuppressive tumor-infiltrating M2-like TAMs are also associated with worse clinical outcome in TNBC [66]. In addition, CAF-S1, a specific subset of cancer-associated fibroblasts (CAFs), has been shown to be enriched in TNBCs and to enhance the capacity of regulatory T cells, which in turn, promote resistance to ICBs [67].

### 4.2. Tumor-Intrinsic Mechanisms of Resistance

Aberrant activation of oncogenic pathways has been associated with an immune-cold TME. A multi-omics profiling with samples from TNBC patients has proposed that the TNBC microenvironment phenotypes could be classified into three clusters [68]. The “immune-inflamed” cluster 3 showed high immune cell infiltration and immune checkpoint molecule expression, whereas the “innate-immune inactivated” cluster 2 displaying higher mutation rates in the PI3K-AKT pathway was characterized by inactivated innate immune cell and non-immune stromal cell infiltration [68]. The “immune-desert” cluster 1 with nearly no infiltrating immune cells tended to harbor *MYC* copy number amplification. MYC has been shown to orchestrate the poor immunogenic TME of TNBC [69]. Mechanistically, MYC transcriptionally upregulates *DNMT1*, a DNA methyltransferase that downregulates the cyclic GMP-AMP synthase-stimulator of interferon genes (cGAS-STING) pathway [69]. Combinatorial treatment of PD-1 inhibitors with decitabine, a DNA methyltransferase inhibitor, elicited the profound anti-tumor effect in MYC-overexpressed TNBC [69]. Another study also demonstrated that STING agonists orchestrate the inflamed TME and promote CD8+ T cell-mediated anti-tumor immunity in TNBC [70]. Activation of the Ras-mitogen-activated protein kinase (MAPK) pathway is also associated with a reduction of MHC, PD-L1, and TIL in TNBC, and consequently, MEK inhibitors in combination with anti-PD1 antibodies synergistically suppressed tumor growth in murine models [71].

Novel immune checkpoint molecules have been implicated in an evasion of TNBC from antitumor immunity. Glycosylation of B7-H4, an inhibitory immune checkpoint molecule that is one of the B7 family ligands, enhances its stability and leads to B7-H7 overexpression in immune-cold TNBC, which causes a decrease in immunogenic cell death [72]. Combined treatment with a B7-H4 glycosylation inhibitor with anti-PD-L1 antibody plus camsirubicin effectively reduced tumor growth in mouse TNBC models [72]. In a more recent study, aberrantly glycosylated B7-H3 was shown to promote the immunosuppressive TME of TNBC [73]. Glycosylated B7-H3 by FUT8, a fucosyltransferase, suppressed activity of T cells in the TME and trafficking of NK cells into tumors [73]. 2F-Fuc, a fucosylation inhibitor, synergistically improved the efficacy of anti-PD-L1 therapy in murine TNBC tumor models [73]. In addition, *LINK-A*, a long non-coding RNA, was shown to facilitate proteasome-mediated degradation of components of peptide-loading complex (PLC), orchestrating the formation of stable peptide–MHC1 complexes that promote T cell response [74]. Inhibition of *LINK-A* by locked nucleic acids increased stability of PLC components, sensitizing tumors to ICB treatment [74].

Subsets of TNBC are also able to intrinsically resist T cell-mediated cytotoxicity. Li et al. (2020) reported that Tenascin-C (TNC) is highly expressed in autophagy-deficient TNBC and is responsible for the suppression of T cell-mediated cytotoxicity [75]. Inhibition of TNC using anti-TNC antibodies sensitized autophagy-impaired TNBC to the PD-L1 blockade [75]. A genome-scale CRISPR knockout screen has identified SOX4 and Integrin αv as novel players that confer resistance to cytotoxic T cells [76]. Treatment with integrin αvβ6-blocking antibody resulted in an induction of CD8+ T cell-mediated cytotoxicity [76]. Furthermore, combination of integrin αvβ6-blocking antibody with anti-PD-1 antibody elicited a profound anti-tumor effect in murine TNBC models [76].

Pro- or anti-inflammatory cytokines that originate from cancer cells have also been implicated in the immune evasion of TNBC. Araujo et al. (2018) reported that a high level of CCL5 expression, which shows a positive correlation with the level of TILs, is associated with longer distant recurrent free survival in TNBCs [77]. Analysis of TCGA TNBC samples identified histone lysine-specific demethylase 1 (LSD1) as a novel suppressor of T cell chemoattractants, such as CCL5, CXCL9, and CXCL10 [78]. Moreover, B cell lymphoma 9 (BCL9) and B cell lymphoma 9-like (BCL9L), transcriptional co-activators of β-catenin, were shown to upregulate TGF-β1 and suppress intratumoral infiltration of CD8+ T cells in TNBCs [79]. Inhibition of BCL9/BCL9L combined with anti-PD-L1 antibodies elicited synergistic anti-tumor responses [79]. A large-scale in vivo CRISPR knockout screening identified Cop1, an E3 ubiquitin ligase, as a regulator of M2 macrophage infiltration in murine TNBC [80]. Deletion of Cop1 in cancer cells led to upregulation of C/ebpδ protein stability, which suppresses expression of chemokines involved in macrophage infiltration, thus demonstrating that Cop1 can serve as a target for increasing immunotherapy effectiveness by modulating macrophage infiltration in TNBC [80]. Li et al. (2018) reported that high glycolytic rate of cancer cells supports the development of MDSC via upregulation of G-CSF and GM-CSF, which in turn, promotes the immunosuppressive TME of TNBC [81]. In addition, cancer cell-derived exosomes, which contain various types of biomolecules including proteins and RNAs, have been shown to prevent immunogenic TME in TNBCs. Wen et al. (2016) revealed that exosomes derived from highly metastatic breast cancer cells are taken up by CD45+ bone marrow-derived cells in the common sites for breast cancer metastasis and cause an accumulation of myeloid-derived immunosuppressive cells in the sites [82]. Other studies have demonstrated that Transforming growth factor-β (TGF-β), PD-L1, and circular RNA encapsulated by exosomes attenuate the inflammatory niche and thus likely promote resistance to ICBs [83,84].

Metabolic intermediates have also been identified as an immunosuppressive messenger in TNBC. Lim at al. (2016) reported that epidermal growth factor (EGF)-mediated stimulation of glycolysis increases lactate production, which inhibited cytotoxic activity via suppression of IFNγ and IL-2 expression in T cells [85]. Noonepalle et al. (2017) reported that T cell activation causes an increase in the expression of indoleamine-pyrrole 2,3-dioxygenase 1 (IDO1) in TNBC cells via IFNγ signaling [86]. IDO1 is an enzyme that catalyzes the production of kynurenine, a metabolite suppressing effector T cells. In addition, this notion is further supported by another study demonstrating that inhibition of IDO in combination with ICB synergistically attenuates tumor growth and prolongs overall survival of mice bearing murine TNBC tumors [87].

## 5. Conclusions

ICBs have shown promising results in TNBC treatment. However, a sizable number of patients do not initially respond to ICBs or acquire resistance to it. Currently, various ongoing clinical trials are exploring the clinical benefit of combining ICBs with treatments, including chemotherapy, molecular targeted therapies, and cancer vaccines. Based on the preclinical studies that were reviewed herein, we propose putative combinatorial strategies overcoming resistance to immune checkpoint blockades (Table 2). Nonetheless, further understanding of various factors that influence the anti-tumor immunity could provide insights that lead to identification of predictive biomarkers that may assist in selecting patients that would best benefit from immunotherapy and development of novel strategies that could overcome resistance to immunotherapy.

**Table 2 biomedicines-10-01130-t002:** Putative strategies to overcome resistance to immune checkpoint blockades.

Intervention	Related Mechanism of Resistance/Mode of Action	Stages of Development
Cyclophosphamide	Tumor-extrinsic/activates stem-like CD8+ T cells	Pre-clinical study [63], NCT03164993 (active, atezolizumab + cyclophosphamide + pegylated liposomal doxorubicin), NCT01898117 (recruiting, atezolizumab + carboplatin + cyclophosphamide), NCT03498716 (recruiting, atezolizumab + cyclophosphamide)
Vinorelbine	Tumor-extrinsic/activates stem-like CD8+ T cells	Pre-clinical study [63], NCT03254654 (completed), NCT02555657 (completed), NCT01104259 (completed)
Decitabine	Tumor-intrinsic/increases T cell infiltration in MYC-overexpressing TNBC by activating the STING pathway	Pre-clinical study [69], NCT02957968 (recruiting, neoadjuvant pembrolizumab + decitabine)
Synthetic cyclic dinucleotide	Tumor-intrinsic/activates the STING pathway, promotes CD8+ T cell-mediated anti-tumor immunity	Pre-clinical study [69]
Trametinib	Tumor-intrinsic/upregulates MHC and PD-L1 expression	Pre-clinical study [71], NCT01467310 (completed), NCT01964924 (completed), NCT02900664 (completed), NCT01155453 (completed), NCT01138085 (completed)
Selumetinib	Tumor-intrinsic/upregulates MHC and PD-L1 expression	Pre-clinical study [71], NCT02685657 (status unknown), NCT02583542 (status unknown)
2-fluoro-L-fucose	Tumor-intrinsic/decreases B7H3 glycosylation which sensitizes TNBC cells to anti-PD-L1 therapy	Pre-clinical study [73]
LINK-A locked nucleic acid	Tumor-intrinsic/stabilizes the peptide loading components and sensitizes tumors to ICB	Pre-clinical study [74]
Anti-Tenascin-C antibody	Tumor-intrinsic/sensitizes autophagy-deficient TNBC to anti-PD-1/PD-L1 therapy	Pre-clinical study [75]
Integrin αvβ6-blocking antibody	Tumor-intrinsic/increases CD8+ T cell-mediated cytotoxicity	Pre-clinical study [76]
hsBCL9_CT_-24	Tumor-intrinsic/promotes cytotoxic T cell and dendritic cell infiltration while reducing regulatory T cells	Pre-clinical study [79]

## Figures and Tables

**Figure 1 biomedicines-10-01130-f001:**
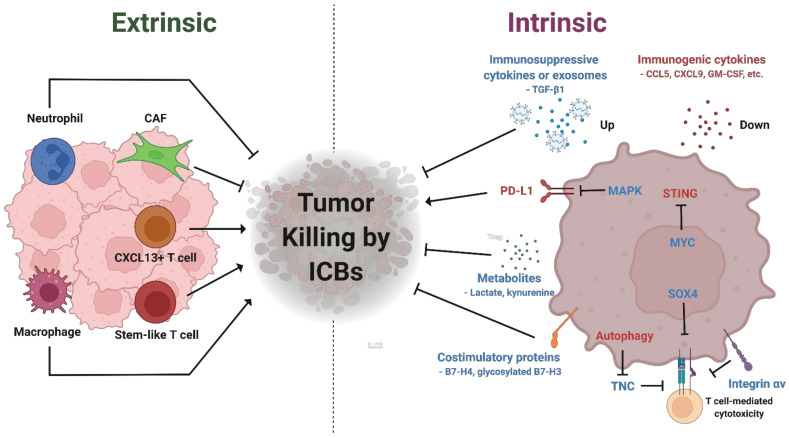
Molecular mechanisms of resistance to ICB in TNBCs. Resistance to ICBs can be driven by a lack of CXCL13+ T cells, stem-like CD8+ T cells, macrophages or the presence of neutrophils, or cancer-associated fibroblasts (CAF). In addition, aberrant activation of MAPK and MYC suppresses the expression of PD-L1 and STING, respectively. SOX4, Integrin αv, and Tenascin-C (TNC) overexpression caused by autophagy deficiency suppresses T cell-mediated cytotoxicity. Overexpression or chemical modification of costimulatory proteins and excessive immunosuppressive metabolites promote ICB resistance. This figure was created with Biorender.com (accessed on 22 October 2021).

**Table 1 biomedicines-10-01130-t001:** Clinical trials investigating the effect of combinatorial treatment regimens of PD-1/PD-L1 inhibitors with chemotherapy in TNBC.

NCT Number	Other IDs/Acronyms	Status	Interventions
NCT02768701	LCCC 1525	Active, not recruiting	Pembrolizumab + Cyclophosphamide
NCT03121352	CASE6115	Active, not recruiting	Carboplatin + Nab-paclitaxel + Pembrolizumab
NCT03567720	OMS-I141, KEYNOTE-890, MK3475-890	Recruiting	Tavokinogene telseplasmid + Pembrolizumab + Immunopulse + Nab-paclitaxel
NCT04191135	7339-009, 2019-001892-35, MK-7339-009, KEYLYNK-009, 195082	Active, not recruiting	Pembrolizumab + Carboplatin + Gemcitabine
NCT02734290	16-001	Active, not recruiting	Pembrolizumab + Paclitaxel, Pembrolizumab + Capecitabine
NCT03639948	NeoPACT, IIT-2017-NeoPACT	Recruiting	Carboplatin + Docetaxel + Pembrolizumab + Pegfilgrastim
NCT05174832	COMPLEMENT	Not yet recruiting	Cisplatin + Nab-paclitaxel + Pembrolizumab, Cisplatin + Nab-paclitaxel + Pembrolizumab + Olaparib
NCT02513472	ENHANCE-1	Completed	Eribulin Mesylate + Pembrolizumab
NCT02622074	3475-173, 2015-002405-11, MK-3475-173, KEYNOTE-173	Completed	Pembrolizumab + Nab-paclitaxel + Anthracycline (doxorubicin) + Cyclophosphamide, Pembrolizumab + Nab-paclitaxel + Anthracycline (doxorubicin) + Cyclophosphamide + Carboplatin, Pembrolizumab + Anthracycline (doxorubicin) + Cyclophosphamide + Paclitaxel
NCT02755272	BR-076, 16-1013	Recruiting	Pembrolizumab + Carboplatin + Gemcitabine, Carboplatin + Gemcitabine
NCT02819518	3475-355, 2016-001432-35, 163422, MK-3475-355, KEYNOTE-355	Active, not recruiting	Pembrolizumab + Nab-paclitaxel, Pembrolizumab + Paclitaxel, Pembrolizumab + Gemcitabine + Carboplatin, Pembrolizumab + Nab-paclitaxel + Paclitaxel + Gemcitabine + Carboplatin, Nab-paclitaxel + Paclitaxel + Gemcitabine + Carboplatin + Normale Saline Solution
NCT03752723	GX-I7-CA-006	Recruiting	GX-I7 + Pembrolizumab + Cyclophosphamide, GX-I7 + Pembrolizumab
NCT03036488	3475-522, 2016-004740-11, 173567, MK-3475-522, KEYNOTE-522	Active, not recruiting	Pembrolizumab + Carboplatin + Paclitaxel + Doxorubicin + Epirubicin + Cyclophosphamide + Granulocyte colony stimulating factor, Carboplatin + Paclitaxel + Doxorubicin + Epirubicin + Cyclophosphamide + Placebo + Granulocyte colony stimulating factor
NCT05112536	G1T28-212	Recruiting	Trilaciclib + Cylophosphamide + Doxorubicin + Paclitaxel + Carboplatin (Investigator discretion) + Pembrolizumab (Investigator discretion)
NCT03289819	NIB, IFG-NIB-01, 2016-003102-14, U1111-1188-3915	Completed	Pembrolizumab + Nab-paclitaxel + Epirubicin + Cyclophosphamide
NCT04443348	20-157	Recruiting	Pembrolizumab + Paclitaxel + Carboplatin + Cyclophosphamide + Doxorubicin + Capecitabine, Radiation Therapy Boost + Pembrolizumab + Paclitaxel + Carboplatin + Cyclophosphamide + Doxorubicin + Capecitabine
NCT04265872	020-008	Recruiting	Bortezomib + Pembrolizumab + Cisplatin
NCT03396445	5890-001, MK-5890-001	Recruiting	MK-5890, MK-5890 + Pembrolizumab, MK-5890 + Pembrolizumab + Pemetrexed + Carboplatin, MK-5890 + Pembrolizumab + Nab-paclitaxel
NCT05093387	NU 19B07, NCI-2021-09317, STU00212682	Not yet recruiting	Carboplatin + Pembrolizumab + Transferrin Receptor-Targeted Liposomal p53 cDNA
NCT03044730	NU 16B08, STU00203215	Active, not recruiting	Capecitabine + Pembrolizumab
NCT02648477	15295, NCI-2015-02194	Active, not recruiting	Doxorubicin Hydrochloride + Pembrolizumab, Anastrozole + Exemestane + Letrozole + Pembrolizumab
NCT05007106	7684A-005, MK-7684A-005, jRCT2031210335, 2021-001700-15	Recruiting	Pembrolizumab/Vibostolimab Co-Formulation, Pembrolizumab, Pembrolizumab/Vibostolimab Co-Formulation + Lenvatinib, Pembrolizumab/Vibostolimab Co-Formulation + 5-Fluorouracil + Cisplatin, Pembrolizumab/Vibostolimab Co-Formulation + Paclitaxel
NCT03213041	NU 16B14, NCI-2017-00330	Recruiting	Carboplatin + Pembrolizumab
NCT04060342	KEYNOTE-A36, GB1275-1101	Active, not recruiting	GB1275, GB1275 + Pembrolizumab, GB1275 + Nab-paclitaxel + Gemcitabine
NCT02957968	MCC-15-11083, NCI-2016-01980	Recruiting	Doxorubicin + Cyclophosphamide + Paclitaxel + Carboplatin + Decitabine + Pembrolizumab
NCT05177796	2020-0715, TN-IBC	Not yet recruiting	Panitumumab + Pembrolizumab + Paclitaxel + Carboplatin + Doxorubicin Hydrochloride + Cyclophosphamide
NCT05069935	FT538-102	Not yet recruiting	FT538 + Cyclophosphamide + Fludarabine + either avelumab, atezolizumab, nivolumab, or pembrolizumab
NCT04954599	TUMAGNOSTIC, 2021-000423-12, 694812	Not yet recruiting	CP-506, CP-506 + Carboplatin, CP-506 + Immune checkpoint inhibitor
NCT04148911	EL1SSAR, MO39874, 2019-002488-91	Recruiting	Atezolizumab + Nab-Paclitaxel
NCT03125902	MO39196, 2016-004024-29, IMpassion131	Active, not recruiting	Atezolizumab + Paclitaxel, Placebo + Paclitaxel
NCT02425891	WO29522, 2014-005490-37	Completed	Atezolizumab + Paclitaxel, Placebo + Paclitaxel
NCT03164993	ALICE, ML39079_ALICE	Recruiting	Pegylated liposomal doxorubicin + Cyclophosphamide + Atezolizumab, Pegylated liposomal doxorubicin + Cyclophosphamide + Placebo
NCT03498716	IMpassion030, WO39391, 2016-003695-47, BIG 16-05, AFT-27, ALEXANDRA	Recruiting	Atezolizumab + Paclitaxel + Dose-dense Doxorubicin or dose-dense Epirubicin + Cyclophosphamide, Paclitaxel + Dose-dense Doxorubicin or dose-dense Epirubicin + Cyclophosphamide
NCT04584112	CO42177, 2020-000531-47	Active, not recruiting	Tiragolumab + Atezolizumab + Nab-paclitaxel, Tiragolumab + Atezolizumab + Nab-paclitaxel + Carboplatin + Doxorubicin + Cyclophosphamide + Granulocyte colony-stimulating factor (G-CSF) + Granulocyte-macrophage colony-stimulating factor (GM-CSF), Tiragolumab + Atezolizumab + Nab-paclitaxel + Doxorubicin + Cyclophosphamide + G-CSF + GM-CSF
NCT04739670	BELLA, 19/002	Not yet recruiting	Atezolizumab + Bevacizumab + Gemcitabine + Carboplatin
NCT04177108	CO41101, 2019-000810-12	Active, not recruiting	Atezolizumab + Ipatasertib + Paclitaxel, Placebo for Atezolizumab + Ipatasertib + Paclitaxel, Placebo for Atezolizumab + Placebo for Ipatasertib + Paclitaxel, Atezolizumab + Paclitaxel + Placebo for Ipatasertib
NCT03197935	IMpassion031, WO39392, 2016-004734-22	Active, not recruiting	Atezolizumab + Nab-paclitaxel + Doxorubicin + Cyclophosphamide + Filgrastim + Pegfilgrastim, Placebo + Nab-paclitaxel + Doxorubicin + Cyclophosphamide + Filgrastim + Pegfilgrastim
NCT03371017	IMpassion132, MO39193, 2016-005119-42	Recruiting	Atezolizumab + Gemcitabine + Capecitabine + Carboplatin, Placebo + Gemcitabine + Capecitabine + Carboplatin
NCT04770272	neoMono, Phaon1	Recruiting	Atezolizumab + Carboplatin + Paclitaxel + Epirubicin + Cyclophosphamide
NCT02530489	2014-1043, NCI-2015-01537, 2014-1043	Active, not recruiting	Atezolizumab + Nab-paclitaxel
NCT03206203	VICC BRE 15136, NCI-2017-01150	Active, not recruiting	Atezolizumab + Carboplatin
NCT03756298	ATOX-2018	Recruiting	Atezolizumab + Capecitabine, Capecitabine
NCT04408118	ATRACTIB, MedOPP150, 2019-001503-20	Recruiting	Atezolizumab + Paclitaxel + Bevacizumab
NCT01898117	Triple-B	Recruiting	Carboplatin/Cyclophosphamide, Carboplatin/Cyclophosphamide + Atezolizumab, Paclitaxel, Paclitaxel + Atezolizumab
NCT02322814	M13TNB, 2013-001484-23, NL44403.031.13	Completed	Drug: Cobimetinib|Drug: Paclitaxel|Drug: Placebo|Drug: Atezolizumab|Drug: Nab-Paclitaxel
NCT02883062	NCI-2016-01301, NCI-2016-01301, 201706104, 10013	Active, not recruiting	Carboplatin + Paclitaxel, Atezolizumab + Carboplatin + Paclitaxel
NCT03800836	CO40151, 2017-001957-15	Active, not recruiting	Ipatasertib + Paclitaxel + Atezolizumab, Ipatasertib + Nab-paclitaxel + Atezolizumab, Ipatasertib + Atezolizumab, Ipatasertib + Paclitaxel + Atezolizumab + Doxorubicin and Cyclophosphamide
NCT04849364	PERSEVERE, HCRN BRE18-334	Recruiting	Capecitabine + Talazoparib + Atezolizumab + Inavolisib
NCT03961698	MARIO-3, IPI-549-03	Recruiting	IPI-549 + Atezolizumab + Nab-paclitaxel
NCT04639245	RG1007463, NCI-2020-06602, 10420	Recruiting	Atezolizumab + Cyclophosphamide + Fludarabine + MAGE-A1-specific T Cell Receptor-transduced Autologous T-cells + PD1 Inhibitor
NCT03424005	Morpheus-TNBC, CO40115, 2017-002038-21	Recruiting	Atezolizumab + Nab-paclitaxel, Atezolizumab + Tocilizumab + Nab-paclitaxel, Atezolizumab + Sacituzumab Govitecan, Capecitabine, Atezolizumab + Ipatasertib, Atezolizumab + SGN-LIV1A, Atezolizumab + Bevacizumab + Selicrelumab, Atezolizumab + Chemotherapy (Gemcitabine + Carboplatin or Eribulin)

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
