# Peer review of "Immune Checkpoint Blockades in Triple-Negative Breast Cancer: Current State and Molecular Mechanisms of Resistance"

_biomedicines, 2022, doi:10.3390/biomedicines10051130_

Round 1
Reviewer 1 Report
The authors of the present review article entitled “Molecular mechanisms of resistance to immune-checkpoint blockade in triple-negative breast cancer” cover the wide aspects of resistance mechanisms against immunotherapies in triple-negative breast cancer.
The authors provide an in-depth and up-to-date discussion about the various resistance mechanisms in TNBC and also include the various ongoing clinical trials which is commendable.
The manuscript is straightforward, well written, concise within the scope of MDPI- Biomedicines.
However, I have a few minor concerns/suggestions that should be addressed before publishing this work:
Suggestion/s:
- The authors did a nice job in writing this manuscript. However, they should in brief mention what could/should/can be done to overcome resistance to immune-checkpoint blockade in triple-negative breast cancer.
- For example, modulating epigenetic factors?
- Overcoming T-cell exhaustion.
- Improving TME?
- The role of Gut microbiota and modulating it etc.
Reviewer 2 Report
The review covers the current immune checkpoint blockade clinical trials and predictive biomarkers of immunotherapy in TNBCs, with only a third of the manuscript focusing on putative molecular mechanisms of ITB resistance in TNBC. It should be noted that the review is titled “Molecular mechanisms of resistance to immune-checkpoint blockade in triple-negative breast cancer”. In this regard, I propose to change the title so that it corresponds not to the part, but to the entire manuscript.
My second and fundamental remark is the absent of mention of exosomes, which are one of the main players in the spread of resistance to immune checkpoint blockade. Authors need to familiarize themselves with the current literature and significantly rewrite Section 4.
However, in general, the manuscript is written in good language, included 1 figure and 1 table.
I think that manuscript entitled "Molecular mechanisms of resistance to immune-checkpoint blockade in triple-negative breast cancer" should be accepted for publication in Biomedicines after major revision.
Round 2
Reviewer 2 Report
The manuscript has been successfully modified and recommended for publication.